# Independent control of amplitude and period in a synthetic oscillator circuit with modified repressilator

Fengyu Zhang [1,2], Yanhong Sun[2], Yihao Zhang [1,3], Wenting Shen[2,3], Shujing Wang[2,3], Qi Ouyang[3] & Chunxiong Luo [2,3,4,5 ✉]

Synthetic Biology aims to create predictable biological circuits and fully operational biological systems. Although there are methods to create more stable oscillators, such as repressilators, independently controlling the oscillation of reporter genes in terms of their amplitude and period is only on theoretical level. Here, we introduce a new oscillator circuit that can be independently controlled by two inducers in *Escherichia coli*. Some control components, including σECF11 and NahR, were added to the circuit. By systematically tuning the concentration of the inducers, salicylate and IPTG, the amplitude and period can be modulated independently. Furthermore, we constructed a quantitative model to forecast the regulation results. Under the guidance of the model, the expected oscillation can be regulated by choosing the proper concentration combinations of inducers. In summary, our work achieved independent control of the oscillator circuit, which allows the oscillator to be modularized and used in more complex circuit designs.

[1] School of Life Sciences and Peking-Tsinghua Center for Life Sciences, Peking University, 100871 Beijing, China. [2] The State Key Laboratory for Artificial Microstructures and Mesoscopic Physics, School of Physics, Peking University, Beijing, China. [3] Center for Quantitative Biology, Academy for Advanced Interdisciplinary Studies, Peking University, Beijing, China. [4] Wenzhou Institute, University of Chinese Academy of Sciences, Wenzhou, Zhejiang, China. [5] Oujiang Laboratory, Wenzhou, Zhejiang, China. ✉email: pkuluocx@pku.edu.cn

Many important physiological processes are controlled by oscillatory signals, such as the P53 protein, cell cycles[1,2] and biological circadian rhythms[3,4]. Via complex control circuits[5], organisms can achieve stable oscillations and adjust the correlation parameters—period, amplitude, and phase—accurately. Compared to these natural cyclic phenomena, synthetic oscillations are difficult to control in such complex networks for now; thus, stable and accurate control of oscillation is a problem. It is needed to be emphasized synthetic circuit with the independent control of amplitude and period is important because of its wide range of applications. The precise control can be quite useful for frequency analysis of downstream networks, frequency encoding and pulse-based signal processing. A modularized regulatable synthetic oscillation circuit can also been used in the design of more complex gene circuits, such as two-dimensional biosensors who need distinct responses from amplitude and period, periodic administration of therapeutic molecules[6].

Recently, synthetic biology has sought to make oscillations robust and tunable in two kinds of synthetic oscillators: the dual-feedback oscillator (DFO)[7] and repressilator (RLT)[8]. For the DFO, some amount of control has been achieved using two changed inducer concentrations[9] or extra control components[10]. However, the independent control of the RLT, which was developed earlier and more widely researched than the DFO, has still not been realized. Based on its general principle, that is, a negative feedback loop with a time delay[11], the RLT is constructed of three transcriptional repressors acting in sequence[9]. The three transcriptional repressors contain TetR from the Tn10 transposon, cI from bacteriophage λ and LacI from the lactose operon. In a period of oscillation, each repressor inhibits the transcription of the next one, which gives a time delay and leads to the oscillation. However, synthetic oscillation circuits, such as the synthetic RLT, unlike which the control of period and amplitude had been realized in the natural world, was difficult to control the oscillation in the absence for now. Two recent studies have reported significant progress: improvement of stability[12] and the theoretical direction of independent control[6]. These works set the stage for independent controlling RLT experimentally.

To solve the problem of independent regulation, we used the strategy of a dual-input promoter[13] to accomplish independent control of amplitude and period experimentally referring to Tomazou's theoretical work[11]. Two kinds of inducers were used to control the reporting protein node and one oscillation node independently. The accuracy of the control was demonstrated using a microfluidic system[14], and quantitative analysis models were established. Through our proposed quantitative model, the oscillations under a particular combination of inducer concentrations can be predicted; thus, we can choose appropriate inducer combinations to achieve the expected effect. Our work developed the theory of the independent control of the RLT in practice, which may give some guidance to realize an adjustable RLT experimentally and make some progress in the quantification and modularization of RLT.

## Results

**Constructing the gene circuits**. To accomplish the independent regulation of amplitude and period, we used two plasmids to construct the circuit. These two plasmids contained the main body of the RLT and the amplitude regulation system, respectively, which were rebuilt from the pLPT119 and pLPT41 plasmids used in Potvin's work[12] (Supplementary Fig. 1).

To the first plasmid, the most important step was the introduction of the dual-input promoter. This dual-input promoter was developed from the promoter of σECF11[15], which can be activated by σECF11. According to the former work of our group[13], this promoter has insulated promoter cores, and the mutations which avoid these promoter cores will not affect the original function of the promoter. Thus we added a TetR binding site surrounding these promoter cores so that the promoter can be repressed by TetR at the same time. To change the promoter of the reporter gene, mVenus, the original pLTetO-1 promoter was separated and replaced by the dual-input promoter in a subclone. Then the fragment after the replacement was integrated into the original plasmid by Gibson assembly.

As for the other plasmid, we designed the σECF11 gene to be regulated by an NahR-pSal system[16,17]. The σECF11 gene, the NahR gene and relevant promoters were all constructed into the pLPT41 plasmid by Gibson assembly (Fig. 1a and Supplementary Fig. 1).

Through the reconstructions mentioned above, we got the circuit to try to realize the independent regulation of the repressilation. The regulation of the amplitude worked as the following steps. Firstly, σECF11 could bind to the promoter of the reporter gene and activated the expression. When the inducer salicylate ($I_1$) was not added, the NahR gene would express so that the expression of σECF11 would be repressed by NahR. As the rise of the salicylate concentration, the repression of NahR would be weakened and the expression of σECF11 would improve, which would lead to a regulation of amplitude. As for the period, we used IPTG ($I_2$) as the input, which can weaken the repression of LacI. Thus the status switching will be slower but the oscillation can still keep within limits, which lead to a regulation of period (Fig. 1b).

**Constructing the observation system**. To observe an oscillation at the single-cell level, we used the "mother machine" microfluidic system, which consists of a series of growth channels and corresponding main channels in which the growth medium passed at a constant rate[14]. We designed eight main channels on a chip, and each channel had 120 growth channels (Fig. 1c). The main channels were designed to be 100 μm width and 20 μm height. The growth channels were designed to be about 1.5 μm width, 1.5 μm height, and 30 μm in length. The growth channels were on the same side of the main channels which could easily load the bacteria through centrifugal force. Using this device, eight different combinations of inducer concentrations could be preformed in one experiment. In the mother machine, cells were trapped and grew in the growth channel with the continuously replacement of medium in the main channel (Supplementary Fig. 2). The medium was pumped at a speed of 40 μl/h by a syringe pump, which is an appropriate speed to guarantee the nutritional supply for the growth of bacteria and stability of the chip.

**Observing and tuning the oscillation**. To check the results of our design, the *E. coli* cells contained in the synthetic gene circuit were analyzed in the mother machine microfluidic system. An on/off test of the regulation of both amplitude and period was performed to check the feasibility of the inducer regulation and stability of the microfluidic system. For the regulation of amplitude, we set the concentration of IPTG to 0 μM and chose two salicylate concentrations, 0 and 200 μM. The results showed that both the low and high concentration groups grew stably and underwent a complete period of oscillation. The average fluorescence intensity amplitude, defined as the difference between a peak and the nearest previous trough, increased from 17.9 to 1963.0 a.u. after the addition of salicylate, which means the regulation of amplitude worked. Two typical channels were shown in

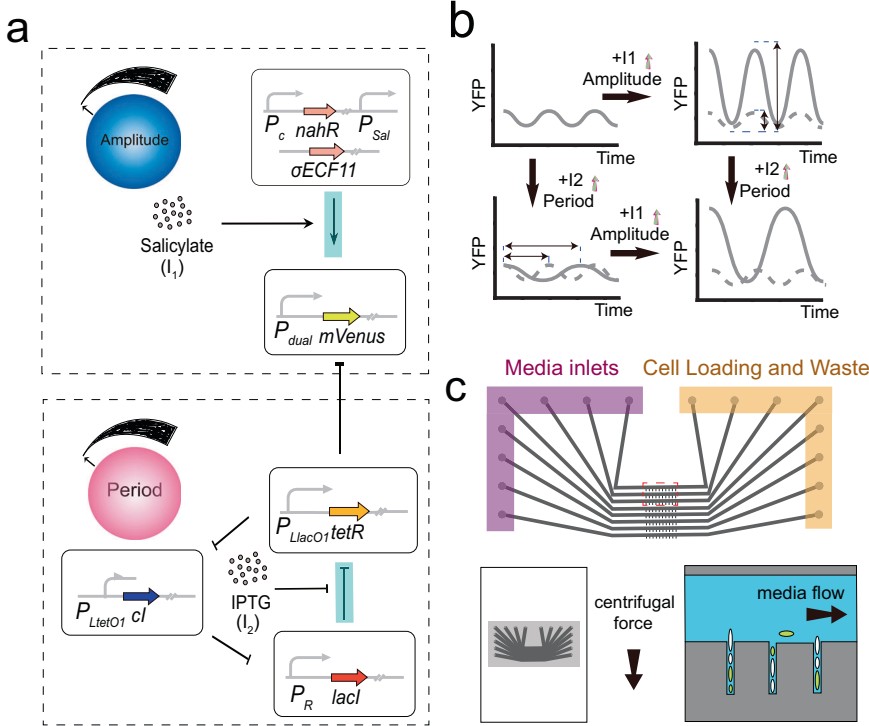

**Fig. 1 Construction and observation of the independent regulation of the repressilator. a** Construction of the gene circuits. The *mVenus* gene, which acts as a reporter, can be regulated by a dual-input promoter. Salicylate can regulate the expression level of the *mVenus* gene, which leads to a change in amplitude. IPTG can regulate the time of the state switch, which leads to a change in period. **b** Schematic diagram of the independent regulation. Input $I_1$ regulates the amplitude and Input $I_2$ regulates the period independently. The combination of $I_1$ and $I_2$ can lead to an oscillation with an expected amplitude and period. **c** Design of the mother machine microfluidic chips to observe the repressilator under different conditions. Eight combinations of different conditions can be observed in an experiment.

Fig. 2a, b, which showed microscopic images of the whole channel over 15 h.

Similarly, for the regulation of the period, we chose two IPTG concentrations, 0 and 10 μM (the concentration of salicylate was set to 100 μM). It could be seen in the two typical channels (Fig. 2c, d) that the 0 μM IPTG concentration group had three peaks and the 10 μM IPTG concentration group had only two peaks, which meant that the period of oscillation increased with the IPTG concentration.

To further confirm the independent regulation ability of the RLT circuit, we performed a series of dose-response experiments. Totally $4 \times 4$ combinations of inducer concentrations were used in the amplitude and period regulation. The concentration of IPTG had a gradient of 0, 2.5, 5, 10 μM and the salicylate had a gradient of 0, 50, 100, 200 μM. The experiments showed the similar results as the on/off test. Five typical fluorescence intensity curves of mother cells for each group were shown in Fig. 3.

The statistical results of the dose-response experiments were shown in Fig. 4. The results were shown in two forms, which represented the independent regulation of amplitude and the independent regulation respectively. Figure 4a represented the independent regulation of amplitude. It can be seen that in all of the four IPTG concentration groups, which were shown as four curves with different colors, the cells that received different concentrations of salicylate had diverse amplitudes, but the periods were similar. Thus, we realized independent regulation of amplitude via $I_1$, salicylate. Similarly, Fig. 4b represented the independent regulation of period. From the results shown in Fig. 4b, we can see that the amplitudes were similar but the periods were diverse among cells that received different concentrations of IPTG, which satisfied our design. From the

results of these experiments, we can state that independent regulation of RLT was realized at the experimental level.

It is also worth mentioning that the independent regulation was not without limits. The results of some additional experiments showed that the amplitude no longer raised with the increase of salicylate when the concentration was greater than 200 μM. And the period no longer raised with the increase of IPTG when the concentration was greater than 10 μM, either. This could be due to the saturation of the inducers. According to our experiments, we can inform that the amplitude range can be controlled from $48.0 \pm 9.4$ a.u. to $1948.8 \pm 16.6$ a.u. and the period range can be controlled from $323.9 \pm 3.3$ min to $383.1 \pm 1.9$ min More detailed data were shown in Supplementary Fig. 3.

**Constructing the quantitative model**. To systematically describe independent regulation in development, we constructed a quantitative model. The model was mainly based on Tomazou's work[6] with some modifications, which contain changing a regulation of an additional node into an inducer and removing the enzymatic degradation. Some equation forms and parameters referred to the former works in our lab[13,17,18], which consist the function of σECF11 and NahR. The model had three parts, as shown in Fig. 5a, which contained the regulation of period, amplitude, and output.

First, gene expression follows a transitional transcription-translation model[6]. The relevant variables are as follows: the molecular number of functional proteins, which means they are already folded ($R_1$ for λcI, $R_2$ for LacI, $R_3$ for TetR, $R_4$ for NahR, $E$ for σECF11, $Y$ for mVenus); molecular number of unfolded proteins ($X_u$, $X = R_1$, $R_2$, $R_3$, $R_4$, $E$, $Y$); molecular number of corresponding mRNAs ($m_x$, $x = R_1$, $R_2$, $R_3$, $R_4$, $E$, $Y$); translation rate constant ($r_x$, $x = R_1$, $R_2$, $R_3$, $R_4$, $E$, $Y$); folding rate constant

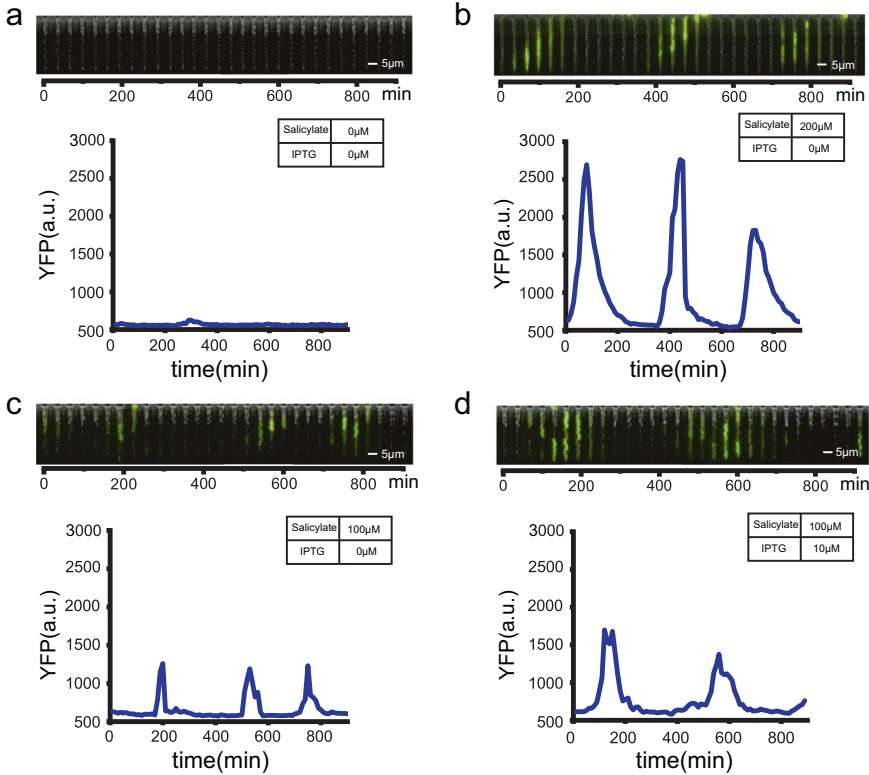

**Fig. 2 Microscope images and the single-cell fluorescence intensity curve of the oscillation in different inducer concentration combinations. a** IPTG, 0 μM; salicylate, 0 μM, **b** IPTG, 0 μM; salicylate, 200 μM, **c** IPTG, 0 μM; salicylate, 100 μM, **d** IPTG, 10 μM; salicylate, 100 μM). Microscope images were merged by the phase contrast and YFP channels. The single-cell curves were drawn from the fluorescence intensity of the mother cell in these channels. Each curve has at least one complete period of oscillation.

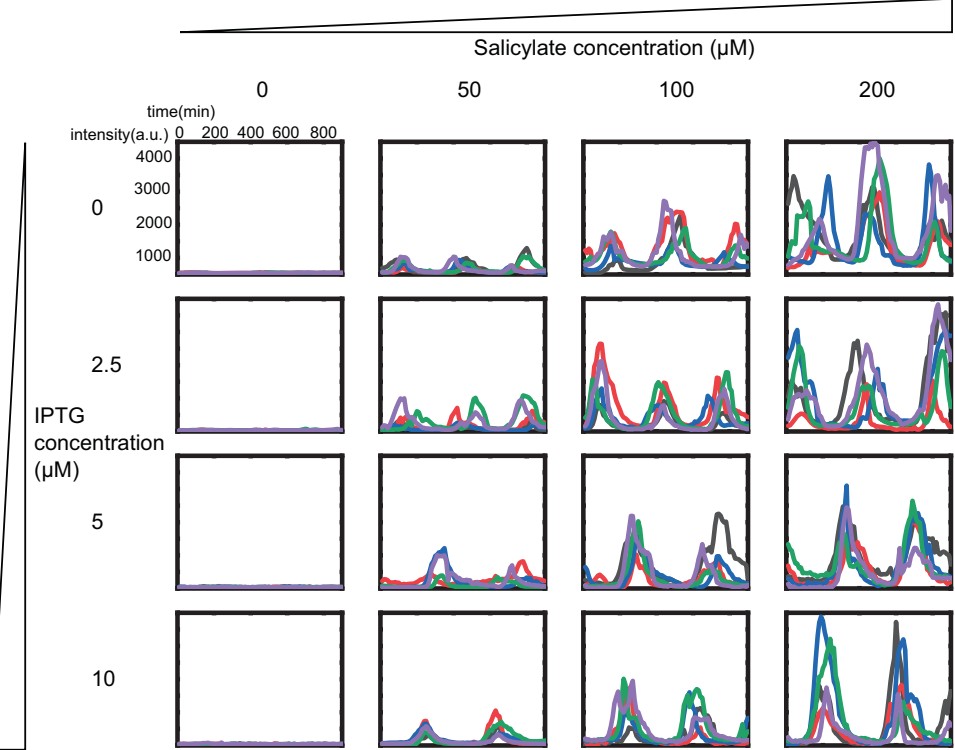

**Fig. 3 Typical curves of the independent regulation of the repressilator circuit.** The concentration of IPTG had a gradient of 0, 2.5, 5, 10 μM and the salicylate had a gradient of 0, 50, 100, 200 μM. Five typical fluorescence intensity curves of mother cells for each group are shown. The curves were re-plot to ensure the alignment of the peaks.

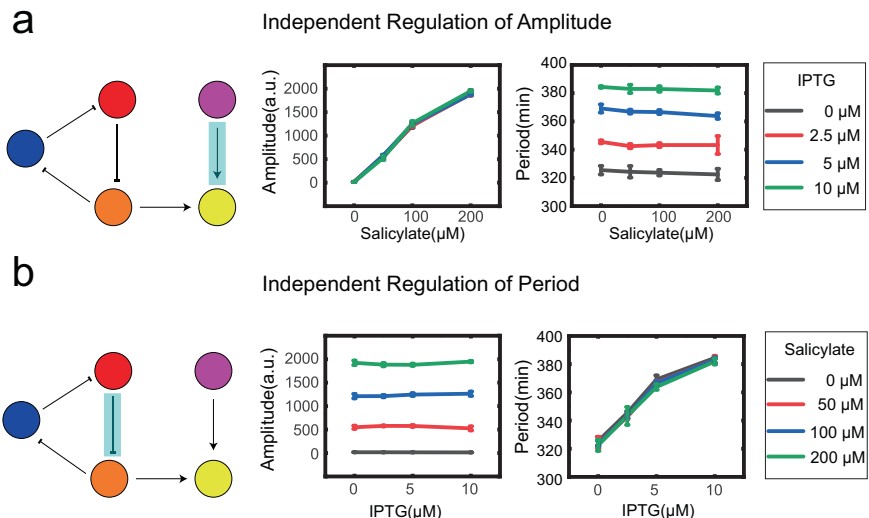

**Fig. 4 Independent regulation of amplitude and period. a** Independent regulation of amplitude. Sixteen groups of single cells were shown as four curves vary with different salicylate concentrations. The four curves respectively represent the IPTG concentration of 0, 2.5, 5, and 10 μM. **b** Independent regulation of amplitude. The same sixteen groups of single cells were shown as four curves vary with different IPTG concentrations. The four curves respectively represent the salicylate concentration of 0, 50, 100, and 200 μM. Each group had the results of three independent experiments and each experiment had over 100 single cells. The data point were calculated from the average of the three experiments and the error bars were calculated from the SD of the average of the three experiments.

$(k_{fx}, x = R_1, R_2, R_3, R_4, E, Y)$; and the protein dilution rate due to cell division ($\delta$). Therefore, the equations for proteins are as follows:

$$\frac{dX_u}{dt} = r_x m_x - k_{fx} X_u - \delta X_u \qquad (1)$$

$$\frac{dX}{dt} = k_{fx} X_u - \delta X \qquad (2)$$

$$(x = R_i (i = 1, 2, 3, 4), E, Y)$$

It needs to be noted that the decay of protein is due to cell division ($\delta$) only, which is different from Tomazou's work[6]. In Tomazou's design, to uncouple the degradation of inhibitor proteins and the reporter protein, they need to be degraded by two orthogonal degradation systems. We used the method of removing enzymatic degradation altogether to solve this problem, which is the method used in Potvin's work[12] and is also mentioned in Tomazou's paper on modeling.

For period regulation, we used IPTG to weaken the repression of LacI, this is simpler than the method to add an extra activator to regulate the expression of TetR, which was used in Tomazou's method[6]. The main variables were as follows: the gene copy number ($p_1$, $p_2$, $p_3$); basal transcription rate constant ($b_1$, $b_2$, $b_3$); maximum transcription rate constant ($a_1$, $a_2$, $a_3$); character concentration, which describes the amount of repressor required for half-maximal repression rates ($k_1$, $k_2$, $k_3$); and decay rate of mRNA, which mainly depends on the degradation rate ($\mu$) for $\mu \gg \delta$. The equations for the *mRNAs* are as follows:

$$\frac{dm_{R_1}}{dt} = p_1 \left( b_1 + a_1 \frac{k_3^2}{k_3^2 + R_3^2} \right) - \mu m_{R_1} \qquad (3)$$

$$\frac{dm_{R_2}}{dt} = p_2 \left( b_2 + a_2 \frac{k_1^2}{k_1^2 + R_1^2} \right) - \mu m_{R_2} \qquad (4)$$

$$\frac{dm_{R_3}}{dt} = p_3 \left( b_3 + a_3 \frac{k_2^2}{k_2^2 + R_2^2} f(I_2) \right) - \mu m_{R_3} \qquad (5)$$

Each equation contains part of the Hill function, and all the Hill coefficients were set to 2. Moreover, the *tetR* gene was controlled by a promoter without a DNA loop in our design, so we used a linear form equation to describe the inducer effect[19]:

$$f(I_2) = \begin{cases} 1 & I_2 < I_0 \\ \frac{I_2}{I_0} & I_2 > I_0, \frac{k_2^2}{k_2^2 + R_2^2} f(I_2) < 1 \\ \frac{k_2^2 + R_2^2}{k_2^2} & \frac{k_2^2}{k_2^2 + R_2^2} f(I_2) \geq 1 \end{cases} \qquad (6)$$

where $I_0$ is the characteristic concentration under which the inducer did not work.

As for the regulation of amplitude, we used NahR and σECF11. The corresponding variables were as follows: the gene copy number ($p_4$, $p_e$), basal transcription rate constant ($b_4$, $b_e$), maximum transcription rate constant ($a_4$, $a_e$), and some variables mentioned above ($k_2$, $\mu$). The equations for the mRNAs of NahR and σECF11 are as follows:

$$\frac{dm_{R_4}}{dt} = p_4 (b_4 + a_4) - \mu m_{R_4} \qquad (7)$$

$$\frac{dm_e}{dt} = p_e \left( b_e + a_e \frac{k_4^2}{k_4^2 + g(R_4)^2} \right) - \mu m_e \qquad (8)$$

where $g(R_4)$ is the depressor effect of NahR after adding salicylate. This step can be described as follows:

$$g(R_4) = \frac{k_{R_4}^2}{k_{R_4}^2 + I_1^2} R_4 \qquad (9)$$

where $I_1$ is the concentration of salicylate. Thus, $g(R_4)$ decreases as $I_1$ increases, leading to the increase of $dm_e/dt$ and, finally, an increased amplitude.

Last but not least, we needed an equation to describe the output, which contained the core of the independent regulation, the dual-input promoter. The dual-input promoter used in our work has been well-studied in our lab[13]. The main variables were as follows: the gene copy number ($p_y$), basal transcription rate

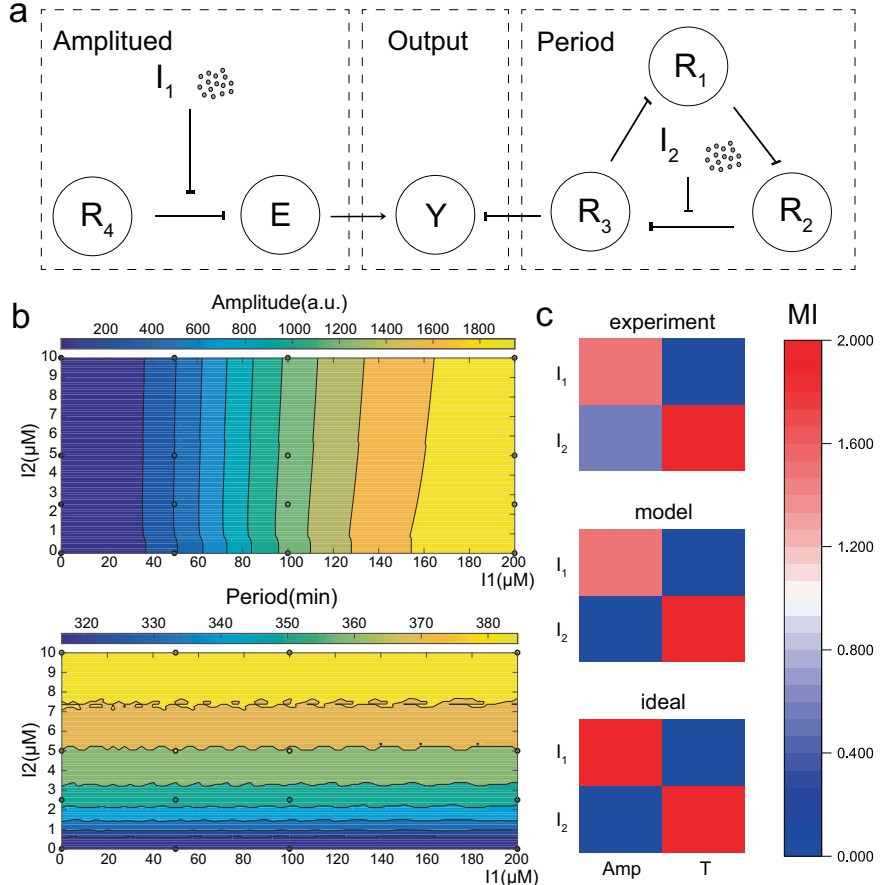

**Fig. 5 Quantitative mathematical modeling of the repressilator. a** Relation diagram of the quantitative model. The model has three parts, the regulation of period, amplitude, and output. **b** Simulation of the quantitative model. The x-axis represents the concentration of salicylate ($I_1$), the y-axis represents the concentration of IPTG ($I_2$). Different values of amplitude and period are represented by different colors. The results of experiments are shown in the form of hollow dots with corresponding colors. **c** MI metrics between the inputs, amplitude, and period. MI (Amp, $T$), MI ($I_2$, Amp), and MI ($I_1$, $T$) are close to zero, meaning that the amplitude and period are not coupled and the regulation of the two inputs are independent.

constant ($b_y$), maximum transcription rate constant ($a_y$), character concentration ($k_e$) and some variables mentioned above ($k_3$, $\mu$). The equation was as follows:

$$\frac{dm_y}{dt} = p_y\left(b_y + a_y \frac{\left(\frac{E}{k_e}\right)^2}{1 + \left(\frac{E}{k_e}\right)^2 + \left(\frac{R_3}{k_3}\right)^2}\right) - \mu m_y \quad (10)$$

Both $E$ and $R_3$ can regulate the transcription of the reporter gene and lead to a change of the fluorescence intensity.

Combining the above equations and the parameters listed in Tables S1 and S2, we can obtain the simulation results, as shown in Fig. 5b, in which the average points of the experiments were also shown. To estimate the degree of predictability of our model We calculated the root mean squared error (RMSE) of our model, which was defined as:

$$\text{RMSE} = \sqrt{\frac{1}{m}\sum_{i=1}^{m}(y_i - \hat{y}_i)^2} \quad (11)$$

The results showed that the RMSE of the amplitudes was 97.0 a.u., and the RMSE of the periods was 4.3 min. These errors are within our acceptable range, so we considered that the model was predictive.

On the other hand, to indicate the orthogonality of the model, we calculated the mutual information (MI) between each of the two inputs/outputs[20]. The MI value between the two variables $x$

and $y$ is defined as:

$$I[X; Y] = \iint p(x,y)\ln\frac{p(x,y)}{p(x)p(y)}dxdy \quad (12)$$

The results were shown in Fig. 5c, which reveals that the regulation is effective, as is the orthogonality.

## Discussion

In this work, we designed and constructed an adjustable RLT gene circuit and realized independent regulation of its oscillation. This is significant progress after the stabilization of the RLT gene circuit. Since synthetic RLT gene circuits have been studied for nearly 20 years, synthetic biologist made great efforts and a lot of progress. At the theoretical level, not only independent regulation[6] but also more complex gene circuits have been designed and simulated[21,22]. Just because of the solid experimental foundation, which means the mature stable regulatory RLT genetic circuit, and the guidance of corresponding theory, we can process this work. As an innovation in practice, our work may give some tips to realize the gene circuit at the experiment level. Our work profited much from Potvin's work[12], which created highly regular and robust oscillations and presented the idea of simply removing existing features. Stability is the basis of regulation. Another important point is the choice of components. Quantifiable gene components are conducive to realizing regulation. Last but not least, it is important to appropriately

simplify the gene circuit without impacting its function. The method we used to regulate the period is to weaken the repression of one node by inducer, which is a simplification compared to the design of an extra activator node used in Tomazou's work[6]. Removing the enzymatic degradation altogether can be seen as a simplification. And it is worth noting that the reporter gene is not the repressor protein itself but regulated by the repressor. This is also a simplification comparing to regulate the repressor protein itself, which is impossible to realize the independent regulation based on existing theories. So it is an effective method to use the original RLT to regulate the period and an additional circuit to regulate the amplitude. Thus we can get the independently regulatable RLT. All these points helped us to complete this work.

Moreover, we also developed a quantitative model to predict the oscillation of the circuit under different inducer concentration combinations. This is an important step for realizing modularization of RLT gene circuits. In previous studies, more attention has been paid to the quality of the oscillation. Therefore, the RLT always acts as the endpoint of control. However, with the use of a quantitative model, we can make the RLT modular. The RLT can be an intermediate node with two inputs and one oscillating output, and this output can regulate other gene circuits that require an oscillating input, such as an oscillating stimulation or activation of gene expression. From the analysis of oscillation, we can obtain the quantitative value of two irrelevant inputs.

Of course, the modular RLT will have many more specific applications such as the frequency analysis, frequency encoding, signal processing and so on, which was also mentioned above. Specially, one broadly important application should be mentioned, that is, pulsatile drug deliver system[23]. Because of the mechanical rhythms of our body, a continuous drug-release system are not ideal at some times, but a pulsatile drug deliver system, which can release the drug periodically. Although there were already some methods to realize the pulsatile drug deliver, they may all have some drawbacks. These kinds of methods were mainly physical methods, such as the coating layers or controlling plugs, need relatively complicated designs and the systems can only realize finite periods. In comparison, the modular RLT can also be used to construct time controlled pulsatile drug system without the problems mentioned above. More importantly, the process of drug synthesis or stimuli generation may also be integrated into the engineering bacteria. So it may be an important future research direction to construct time controlled pulsatile drug deliver systems.

Still, our design may be difficult to apply because of some other couplings. One example is resource constraints, which are always a concern in synthetic biology[24,25]. Competition for shared cellular resources may have some influence on the regulation of oscillation. It is a new challenge for us to explore the limit of the resource constraints and learn how to deal with them. On the other hand, if the RLT is used in the fermentation field as an internal regulatory system, environmental stress may be a problem. It is known that the product of fermentation engineering is always poisonous to the engineered bacteria, especially when products pile up[26]. Therefore, RLT gene circuits are very likely to be influenced in such environments. In this case, exploring methods to avoid this influence, such as by protecting engineered bacteria and separating the product, is also a future research direction.

## Methods

**Stains and plasmids**. *E. coli* Top10 was used for plasmid construction, *E. coli* K-12 DH10B was used for parts characterization, and *E. coli* DHL807 from Potvin's work was used for circuit measuring throughout this study. PCR, DNA ligation and Gibson assembly were used in the construction of the plasmids. The schematic

diagrams of detailed cloning steps were shown in Supplementary Fig. 1 and the plasmids used in this study were listed in Table S3.

**Microfluidic device fabrication**. To fabricate the molds for the microfluidic chip, we used a two-layer photolithography method to create SU8 photoresist (Microchem, Japan) patterns with two heights on the silicon wafer; 6–8 mm thick PDMS was then cast on the silicon wafers and disposed at a curing temperature of 70 °C for 3 h. Then, the PDMS piece was peeled from the silicon wafer, with the shape of the designed structures transferred to the surface of the PDMS piece. After cleaning with Scotch Magic tape (Minnesota Mining and Manufacturing Corporation) and an oxygen plasma treatment, the PDMS piece was bonded to 0.13 mm thick glass and heated overnight at 70 °C. Before cell loading, the chip was degassed in vacuum for 10 min.

**Cell growth and observation**. Bacteria were inoculated from single colonies into a 10 ml glass test tube and then cultured in LB medium at 37 °C in a shaker overnight. Then, the cells were diluted 20-fold with a mixture of 90% M9 medium and 10% LB medium for 3 h. Next, the cells were concentrated for 5 min at 4000 rpm, resuspended in 100 μl of a 0.1% BSA solution and loaded into the microfluidic chip. The microfluidic chip was centrifuged for bacteria loading into the trap lines for 10 min at 4000 rpm, after which the chip was connected to our cultivation-observation device. The cultivation-observation setup contained a Nikon Ti-E inverted fluorescence microscope with an EMCCD camera (Andor iXon×3 DU897) and a CFI plan Apochromat Lambda DM ×60 oil immersion objective (NA 1.40 WD 0.13 mm). The microfluidic chip was placed on the motorized microscope stage (with Encoders), and the incubator system temperature was set to 37 °C. Four micro-syringe pumps were used to inject the medium with a flow rate of 40 μl/h. Images were acquired every 10 min for 24 h or longer. Illumination, exposure time, and camera gain were set to appropriate values and were not changed between experiments.

**Media and buffers**. All the chemicals used in the study were purchased from Sigma-Aldrich unless stated otherwise. LB medium: 10 g/l tryptone, 5 g/l yeast extract, and 10 g/l NaCl. For agar plates, 15 g/l agar was added. M9 medium: 6.8 g/l $Na_2HPO_4$, 3 g/l $KH_2PO_4$, 0.5 g/l NaCl, 1 g/l $NH_4Cl$, 0.34 g/l thiamine, 0.2% casamino acids (BD Biosciences), 0.4% glucose, 2 mM $MgSO_4$, and 100 μM $CaCl_2$. The LB medium for overnight culture contained ampicillin and kanamycin at concentrations of 100 μg/ml to maintain the plasmids. The mixed medium for the microfluidic culture contained ampicillin and kanamycin at concentrations of 1 μg/ml at which the cells can grow normally and the plasmids can be maintained.

**Data analysis and modeling**. For the analysis of microscopic images, we used ImageJ software. Single cells were manually tracked, and the oscillation peaks were found by manually encircling single cells and measuring and comparing the mean fluorescence intensity of each single cell. The periods were calculated by counting the time intervals between two adjacent peaks. The amplitudes were calculated as the difference of the fluorescence intensity between each peak and its nearest trough. The quantitative model was simulated using MATLAB. The forms and the parameters of the equations were mainly from previous work[6,13,17,18] and modified to fit our experiment results. The amplitudes and periods were calculated using the "findpeaks" function.

**Statistics and reproducibility**. All statistics were described in figure legends. Each group of dose-response experiment of the RLT gene circuit was performed three times and the results were expressed as (mean value ± standard deviation). In the comparison among the groups of different inducer concentration combinations, statistical analyses of data for comparison were carried out by Student's *t* test, *p* values < 0.05 were considered to be significant.

**Reporting summary**. Further information on research design is available in the Nature Research Reporting Summary linked to this article.

## Data availability

We declare that all relevant data supporting the findings of this study are available within the article and its Supplementary Information Files, Supplementary Data 1 or from the corresponding authors upon request. The publicly available datasets can also be got at https://pan.baidu.com/s/18O3PcG_Qc7exUfyb_vbBqw with the fetch Code 7214.

## Code availability

The code for the quantitative model construction is available in BioModels (MODEL2111260001).

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

## Acknowledgements

This study was supported the National Key Research and Development Project (2018YFA0900700, 2020YFA0906900, 2021YFF1200500) and the NSFC of China (11974002, 11674010).

## Author contributions

Q.O. and C.L. conceived and supervised the project. F.Z. and C.L. designed the experiment. F.Z., Y.Z., and S.W. performed the experiment. F.Z., Y.S., and W.S. analyzed the data. F.Z. and C.L. wrote the manuscript.

## Competing interests

The authors declare no competing interests.
