## [Transparent Peer Review File · Communications Biology]

Reviewers' comments:

Reviewer #1 (Remarks to the Author):

The short article by Zhang et al reports a variant of the famous repressilator circuit endowed now with two additional genetic connections through which the amplitude and the period of the in vivo oscillations can be externally controlled. The article is well written, the logic of the work is sound, the experiments are well done (with some issues, see below) and the conclusions are basically credible. Yet, these points should be addressed.

1. I do not agree with the claim that the new device has an *orthogonal* control of the amplitude/period. In the best case, the improved repressilator enables an exogenous tuning of these parameters but by no means orthogonal. Authors should refrain from abusing the SynBio jargon.
2. It would be desirable to have separate data and parameters on the input/output function of the NahR/Psal->sEFC11 device as compared to the other (LacI/PLlacO1) to pre-assess their compatibility rather than just showing the final outcome in Fig. 2. The reader is otherwise left wondering about the choice of such specific devices and pre-set parameters vs others and whether the successful performance shown was just a stroke of luck. I am particularly intrigued by the choice of sEFC11 as one of the ingredients of the system, as changes in the balance of sigma factors have major consequences on cell's physiology.
3. In connection to #2 above, it is unusual that a mathematical model follows the actual experiments and not vice versa. Please explain.
4. The key results shown in Fig. 2 are nice, but not Earth-shattering. There seems to be a considerable phenotypic variability in the populations C and D. I wonder whether things would improve if the circuit had been engineered in the chromosome rather than in plasmids.
5. Also in the same Figure 2: it would be of great interest to see for how long the circuit could be running much beyond 800 mins.
6. Authors mention on passing (Discussion, line 303 and ss) the issue of resource allocation, but they provide neither data on it nor any measurement of the physiological impact of the new repressilator in E. coli cells. It would be interesting to just monitor growth in a culture of the engineered strain under no-induction, separate inducers and both inducers. In fact, that there is a (negative?) influence argues against the orthogonality claims indicated above.

Reviewer #2 (Remarks to the Author):

General Comment:

Fengyu Zhang et al. describe the design and characterization of a new repressilator genetic circuit based on the theory described by Potvin. The new circuit design enables controlling the amplitude and periodicity of gene expression independently using two chemical inputs. While the concept and goal of the work are impressive, the focus is a very technical and would be more appropriate for a specialized journal, such as ACS Synthetic Biology.

Specific Comments:

Please expand the introduction to include why tuning the amplitude and periodicity is important besides it hasn't been done before.

Line 57, please add an additional sentence or two describing in more detail how the additional

promoters enable independent control of the repressilator.

Section 'Constructing the gene circuit': Only need a few sentences to explain and the remaining details can be moved into the supplemental.

Line 84, 88, ... 'gibson' should be capitalized to Gibson.

Line 92, Please explain how σ_{ECF11} enables control of the amplitude. This is important for the general reader not familiar with the field.

Line 99, Please explain why the microfluidic system is referred to as 'mother machine'. Also, cite its origin.

Line 105, preform -> perform

Figure 1D can be moved into supplemental. Highlight of the paper is the genetic circuit, not microfluidic design.

Line 129, results in Figure 2A and 2B show that the number of oscillations change. I see one oscillation at ~ 300 minutes for Figure 2A and I see 3 oscillations for Figure 3B, at different times. I thought the oscillation pattern should not change unless IPTG is altered. Can the authors please explain?

Line 138, should 'Figure 2A' be Figure 2C?

Figure 2D. Shouldn't the period (time between cycles change). Seems like the periodicity is within error. How many experiments were ran, and what is the statistical difference?

Figure 2B, 200 μM salicylate doesn't agree data in Figure S2. 200 μM visually has twice the error as the other conditions, but this isn't consistent with Figure 3. Can the authors please explain.

Figure S2. Please move to main text.

Figure S2. Can the authors please explain why IPTG 10 μM has shorter periods compared to 5 μM IPTG. Also, why is this not reflected in Figure 3? There seems to be an inconsistency between Figure S2 and Figure 3 data.

Line 342, the authors mention running the experiment for 24 hrs but the figure data only shows 15 hrs. Can the authors show data for 24 hrs. How stable is the circuit?

The authors should add an additional section describing and demonstrating a broadly important application. This will help demonstrate the broad utility of the newly designed circuit.

Reviewer #3 (Remarks to the Author):

In this paper, Zhang et al. created a new oscillator circuit that can be orthogonally controlled by two inducers in *Escherichia coli*. By systematically tuning the concentration of the inducers, the amplitude and period can be modulated independently. They also constructed a quantitative model to forecast the regulation results. Under the guidance of the model, the expected oscillation can be regulated by

choosing the proper concentration combinations of inducers. This novel design allows the oscillator to be modularized and used in more complex circuit designs, which would be a valuable contribution to the Synthetic Biology field.

The main critique I have is that I think the impact of the project needs to be explained more in detail. In line 52-54, it says. "Because this is a complete synthetic circuit unlike those in the natural world, it is difficult to control the oscillation in the absence of a reference." This statement seems to be the one that justifies the circuit design but needs to be clarified, what is different in this synthetic circuit from natural systems? Why is this needed and important?

Please check for typos:

Fig 4A. It should be "Amplitude".

Fig 1A. If Input I1 regulates the period and Input I2 regulates the amplitude why in the figure the schematic at the top left corner of each box shows the opposite?

Line 158, "and the independent regulation respectively". Please complete the sentence, there's missing information there.

Other comments/questions:

Lines 179-184. Upper limit is mentioned and explained in a nice way there, it may be worth informing the lower limit to complete the range in which the system is functional.

Line 189. Please inform the modifications introduced here to Tomazou's work.

Lines 212-217. Why choosing the approach of removing the enzymatic degradation altogether when in Tomazou's paper the method is mentioned but not used?

For clarification, please elaborate what is different from the other papers mentioned, so that the reader doesn't need to read the other two papers to understand the point made here:

Lines 218-219, "simpler than Tomazou's method", in what way?

Line 276, "Our work profited much from Potvin's work", how?

Lines 279-281, "The method we used to regulate the period is a simplification compared to the design in Tomazou's work.", in what way?

In figure 4C. Why is it that the model predicts better the MI (I1, Amp) rather than MI (I2, Amp)?

Please clarify the following expressions:

Lines 256-258. "It can be seen that the experimental results are generally quantitative and conform to the model, which means the model has a degree of predictability." What does "generally" mean here? I'd expect a reproducible result. Can you estimate the degree of predictability?

Line 130. "The regulation also worked." What kind of regulation and in which sense it did work?

Line 131. "The fluorescence intensity amplitude, defined as the difference between a peak and the nearest previous trough, increased by approximately 100 times, generally after the addition of salicylate". What does "generally" mean here? I'd expect a reproducible result.

Response to Reviewer #1

Reviewer Comments:

Reviewer #1:

General Comments:

The short article by Zhang et al reports a variant of the famous repressilator circuit endowed now with two additional genetic connections through which the amplitude and the period of the in vivo oscillations can be externally controlled. The article is well written, the logic of the work is sound, the experiments are well done (with some issues, see below) and the conclusions are basically credible.

Reply: We thank the reviewer for the thorough and positive assessment of our work on the “The article is well written, the logic of the work is sound, the experiments are well done and the conclusions are basically credible.”. We have revised our manuscript following the reviewer’s suggestions to improve the clarity of the manuscript.

Yet, these points should be addressed.

*1. I do not agree with the claim that the new device has an *orthogonal* control of the amplitude/period. In the best case, the improved repressilator enables an exogenous tuning of these parameters but by no means orthogonal. Authors should refrain from abusing the SynBio jargon.*

Reply: Thanks for the reviewer’s comments. Sorry for our negligence of abusing the SynBio jargon. We have changed “orthogonal” to “independent” to describe our tuning of the amplitude/period. Thank you again for your suggestions to improve the clarity of this paper.

2. It would be desirable to have separate data and parameters on the input/output function of the NahR/Psal->sEFC11 device as compared to the other (LacI/PLlacO1) to pre-assess their compatibility rather than just showing the final outcome in Fig. 2. The reader is otherwise left wondering about the choice of such specific devices and pre-set parameters vs others and whether the successful performance shown was just a stroke of luck. I am particularly intrigued by the choice of sEFC11 as one of the

ingredients of the system, as changes in the balance of sigma factors have major consequences on cell's physiology.

3. In connection to #2 above, it is unusual that a mathematical model follows the actual experiments and not vice versa. Please explain.

Explanation and Modification:

We gratefully appreciate the reviewer's valuable comments and suggestions. We are very sorry for not explaining the detail of our quantitative model construction. Actually the *NahR/PsaI* and $\sigma ECF11$ devices were well studied in former researches in our lab[1][2][3]. The dose-response curve of the separated components were all well measured (Fig. R1). Some authors of this paper also participated these former researches so we can say these results are credible and useful. So this part of our model was developed according to these works including the measuring results and the form of the equations. Besides, the combination of these devices (*NahR/PsaI*, $\sigma ECF11$ and *LacI/Plac*) were used a lot in former researches, that was why we chose them in this work. We have added references to increase the rigor of the article.

Specially, as for the choice $\sigma ECF11$, we can give an explanation as following. Indeed it's well known that changes in the balance of sigma factors have major consequences on cell's physiology, but some kinds of extracytoplasmic function (ECF) sigma factors with their highly conserved promoters were researched to be orthogonal to host sigma factors in *E.coli*[4]. The combination of $\sigma ECF11$ and its promoter is just an excellent one of them, which has high expression level and wide regulation range. Thus we used $\sigma ECF11$ a lot in our former researches and the results were satisfactory. In this work, we still used $\sigma ECF11$ to complete the circuit design and reached to desired results.

Figure R1. Some examples of dose-response curves of the devices used in the design. a. Examples of *NahR/Psal*-salicylate dose-response curves. Reprinted with permission from ref 1. Copyright © 2015 American Chemical Society. b. Examples of $\sigma ECFII/P_{ECFII}$ -input dose-response curves. Reprinted with permission from ref 3. Copyright © 2017, The Author(s).

4. *The key results shown in Fig. 2 are nice, but not Earth-shattering. There seems to be a considerable phenotypic variability in the populations C and D. I wonder whether things would improve if the circuit had been engineered in the chromosome rather than in plasmids.*

Reply: Thanks for the reviewer's suggestion. In fact, we also considered to engineer the circuit in the chromosome previously. But we finally set our goal to build a modularized repressilator circuit which should be conveniently applied into more complex circuit designs. In this case, we thought the circuit is more appropriate to be engineered in plasmids, which can be further designed more easily than in chromosome. Although the results had some phenotypic variability, the data accuracy have already met our requirement and the designability is better, so we prefer to engineer our circuit in plasmids.

5. *Also in the same Figure 2: it would be of great interest to see for how long the circuit could be running much beyond 800 mins.*

Reply: Thanks for the reviewer's insightful comments. Actually we acquired the images for 24h in each experiment. But usually the useful time serials of the data contained less than 24 hours because of the lost of focus after a quite long time photographing in half of the experiments. On the other hand, for the need to re-plot the oscillation curves of single cells to ensure the alignment of the peaks and get more comparable results of single cell(over 100 single cells in each experiments), we only show the data within 900mins. But according to the reviewer's requirement, we provide several examples of 24hours (1440mins) as Fig. R2 which shows relative stable oscillations.

Fig R2. Some examples of the bacteria grow for 24h. Three channels under different condition of inducer concentrations were shown, which also showed relative stable oscillations for longer time.

6. Authors mention on passing (Discussion, line 303 and ss) the issue of resource

allocation, but they provide neither data on it nor any measurement of the physiological impact of the new repressilator in E. coli cells. It would be interesting to just monitor growth in a culture of the engineered strain under no-induction, separate inducers and both inducers. In fact, that there is a (negative?) influence argues against the orthogonality claims indicated above.

Explanation and Modification:

Thanks for the reviewer’s valuable comments and suggestions. We are very sorry for not declaring the influence of the inducers. Actually we also worry about that the inducers may have possible influence to the growth of the cells, especially the rate of cell division, which was an important determinant of oscillation period. So we analyzed the division rates of the cells under no-induction, separate inducers (200µM salicylate or 10µM IPTG) and both inducers. The statistical results were shown in Table.R1. It seems that the inducer concentration used in our work didn’t have any distinct influence to the growth of cells. As for the issue of resource allocation, this was just an supposition based on some results of literature research at the present stage. Although we have already done some calorie restriction experiments (diluted the culture media with PBS to 1/10) but the results showed that both the growth and the oscillation were not influenced at all. Maybe the calorie restriction level was not enough in these experiments. More conditions will be attempted in our future experiments.

Induction conditions	No inducer	200 µ M salicylate only	10 µ M IPTG only	200 µ M salicylate and 10 µ M IPTG
Average division time (min)	34.2	34.6	34.4	34.5
standard deviation	2.4	2.0	2.2	1.6

Table R1. growth contrast of different induction conditions

Thank the reviewer again for all comments and suggestions!

Response to Reviewer #2

Reviewer Comments:

Reviewer #2:

General Comment:

Fengyu Zhang et al. describe the design and characterization of a new repressilator genetic circuit based on the theory described by Potvin. The new circuit design enables controlling the amplitude and periodicity of gene expression independently using two chemical inputs. While the concept and goal of the work are impressive, the focus is a very technical and would be more appropriate for a specialized journal, such as ACS Synthetic Biology.

Reply: We thank the reviewer for the thorough and positive assessment of our work on the “the concept and goal of the work are impressive”. First we want to explain the reason to choose *Communications Biology*. *Communications Biology* is an open access journal from Nature Portfolio publishing high-quality research, reviews and commentary in all areas of the biological sciences, of course including the synthetic biology. And it can be seen *Communications Biology* is great-hearted with the increasing articles and impact factor, so we chose to send our article to *Communications Biology*. Then we have revised our manuscript following the reviewer’s suggestions (see below) to make the manuscript clearer and more rigorous.

Specific Comments:

Please expand the introduction to include why tuning the amplitude and periodicity is important besides it hasn’ t been done before.

Reply: Thanks for the reviewer’s comments and suggestions. We are very sorry for neglecting to introduce the significance of independent regulation. We have expanded the introduction according to the Reviewer’s comments.

“It is needed to be emphasized synthetic circuit with the independent control of amplitude and period is important because of its wide range of applications. The precise control can be quite useful for frequency analysis of downstream networks, frequency encoding and pulse-based signal processing. A modularized regulatable synthetic oscillation circuit can also been used in the design of more complex gene circuits, such as two-dimensional biosensors who need distinct responses from amplitude and period, periodic administration of therapeutic molecules.”

Thank you again for your suggestions to improve the clarity of this paper.

Line 57, please add an additional sentence or two describing in more detail how the additional promoters enable independent control of the repressilator.

Reply: Thanks for the reviewer’s comments and suggestions. We are very sorry for neglecting to introduce the additional promoters in detail. We have added the related instructions according to the Reviewer’s comments.

“This dual-input promoter was developed from the promoter of σ ECF11[15], which can be activated by σ ECF11. According to the former work of our group[13], this promoter has insulated promoter cores, and the mutations which avoid these promoter cores will not affect the original function of the promoter. Thus we added a TetR binding site surrounding these promoter cores so that the promoter can be repressed by TetR at the same time.”

Thank you again for your suggestions to improve the clarity of this paper.

Section ‘Constructing the gene circuit’ : Only need a few sentences to explain and the remaining details can be moved into the supplemental.

Reply: Thanks for the reviewer’s comments and suggestions. We are very sorry for give unnecessary details in the text. We have condensed the ‘Constructing the gene circuit’ section and moved the details into the supplemental according to the Reviewer’s comments. Thank you again for your suggestions to improve the clarity of this paper.

Line 84, 88, ... ‘gibson’ should be capitalized to Gibson.

Reply: Thanks for the reviewer’s careful check. We are very sorry for our negligence of some case mistakes. We have made corrections according to the Reviewer’s comments. Thank you again for your suggestions to improve the clarity of this paper.

Line 92, Please explain how σ ECF11 enables control of the amplitude. This is important for the general reader not familiar with the field.

Reply: Thanks for the reviewer’s comments and suggestions. We are very sorry for neglecting to introduce the function of σ ECF11 in detail. We have added the related instructions according to the Reviewer’s comments.

“The regulation of the amplitude worked as the following steps. Firstly, σ ECF11 can bind to the promoter of the reporter gene and activate the expression. When the inducer salicylate(I₁) was not added, the NahR gene will express so that the expression of σ ECF11 will be repressed by NahR. As the rise of the salicylate concentration, the repression of NahR will be weakened and the expression of σ ECF11 will improve, which lead to a regulation of amplitude.”

Thank you again for your suggestions to improve the clarity of this paper.

Line 99, Please explain why the microfluidic system is referred to as ‘mother machine’. Also, cite its origin.

Reply: Thanks for the reviewer’s comments and suggestions. We have added the related instructions and added references to this part according to the Reviewer’s comments.

“To observe an oscillation at the single cell level, we used the “mother machine” microfluidic system, which consists of a series of growth channels and corresponding main channels in which the growth medium passed at a constant rate[14].”

Thank you again for your suggestions to improve the clarity of this paper.

Line 105, preform -> perform

Reply: Thanks for the reviewer’s careful check. We have made corrections according

to the Reviewer's comments. Thank you again for your suggestions to improve the clarity of this paper.

Figure 1D can be moved into supplemental. Highlight of the paper is the genetic circuit, not microfluidic design.

Reply: Thanks for the reviewer's comments and suggestions. We have moved Figure.1d into the supplemental (Fig. S2) according to the Reviewer's comments. Thank you again for your suggestions to improve the clarity of this paper.

Line 129, results in Figure 2A and 2B show that the number of oscillations change. I see one oscillation at ~300 minutes for Figure 2A and I see 3 oscillations for Figure 2B, at different times. I thought the oscillation pattern should not change unless IPTG is altered. Can the authors please explain?

Explanation and Modification:

Thanks for the reviewer's insightful comments. We also thought the oscillation pattern should not change unless IPTG is altered. Actually there were also 3 oscillations in Figure 2A, which was the same as in Figure 2B. The second peak was at ~600 minutes and the third one was at ~800 minutes which hardly can be found. We are very sorry that the peaks were not conspicuous because the value of the amplitudes were quite small, but this result was consistent with our expectation. More data can be found in the new Figure 3.

Line 138, should 'Figure 2A' be Figure 2C?

Reply: Thanks for the reviewer's careful check. We are very sorry for our negligence of some noting mistakes. We have made corrections according to the Reviewer's comments. Thank you again for your suggestions to improve the clarity of this paper.

Figure 2D. Shouldn't the period time between cycles change. Seems like the periodicity is within error. How many experiments were ran, and what is the statistical difference?

Reply: Thanks for the reviewer's insightful comments. For each combination of

inducer concentrations, we ran 3 repeated experiments and each experiment had over 100 single cells. As for the concentration combination in Figure. 2D, which was 100 μ M salicylate and 10 μ M IPTG, the average of the 3 repeated experiments was 384.0 minutes and the standard deviation was 1.9 minutes. The statistical information was provided in the legend of new Figure 4. This standard deviation value was small enough comparing to the deviations between groups with different IPTG concentrations so we believe that the period was regulated by changing the IPTG concentration.

Figure 2B, 200 uM salicylate doesn't agree data in Figure S2. 200 uM visually has twice the error as the other conditions, but this isn't consistent with Figure 3. Can the authors please explain.

Explanation and Modification:

Thanks for the reviewer's insightful comments. First, we are very sorry that the typical channel shown in Figure 2B was not representative enough. Actually the 200 μ M salicylate typical channel shown in Figure 2B did agree the data in Figure S2(Now Figure 3), but it was one with relative lower amplitude in the group. Thus we changed the example in Figure 2B to a more representative one, whose amplitude was more close to the mean level.

Then we want to explain about the error. Actually the experiment of 200 μ M had bigger error than other conditions, which can also seen in Figure S2(Now Figure 3) and Figure 3(Now Figure 4). But the error is relative small because the data shown in Figure 3(Now Figure 4) was the statistical results of 3 repeat experiments, each experiments is average from more than 100 cells. The single cell results were averaged out in the pre-analysis of each experiment so the statistical error in Figure 3(Now Figure 4) is much smaller than the behaviors of single cell curves.

Figure S2. Please move to main text.

Reply: Thanks for the reviewer's comments and suggestions. We are very sorry for not put the important content in the main text. We have moved Figure S2 into the main text as Figure 3 according to the Reviewer's comments. Thank you again for your

suggestions to improve the clarity of this paper.

Figure S2. Can the authors please explain why IPTG 10 μ M has shorter periods compared to 5 μ M IPTG. Also, why is this not reflected in Figure 3? There seems to be an inconsistency between Figure S2 and Figure 3 data.

Reply: Thanks for the reviewer's insightful comments. Maybe it seems that the IPTG 10 μ M has shorter periods compared to 5 μ M IPTG but that was not as the matter of fact. We calculated the average of the periods shown in Figure S2. The average period of IPTG 5 μ M groups was 362.5 minutes and the average period of IPTG 10 μ M groups was 383.5 minutes, which was consistent with Figure 3(Now Figure 4). We think that maybe the first peaks shown in IPTG 10 μ M groups were earlier than those in IPTG 5 μ M groups so that the periods seemed to be shorter. Of course 20 minutes is not a long time to be distinguished clearly in the graph so the statistical result may be more precise.

Line 342, the authors mention running the experiment for 24 hrs but the figure data only shows 15 hrs. Can the authors show data for 24 hrs. How stable is the circuit?

Reply: Thanks for the reviewer's insightful comments. Actually we acquired the images for 24h in each experiment. But usually the useful time serials of the data contained less than 24 hours because of the lost of focus after a quite long time photographing in half of the experiments. On the other hand, for the need to re-plot the oscillation curves of single cells to ensure the alignment of the peaks and get more comparable results of single cell(over 100 single cells in each experiments), we only show the data within 900mins. But according to the reviewer's requirement, we provide several examples of 24hours (1440mins) as Fig. R3 which shows relative stable oscillations.

Fig R3. Some examples of the bacteria grow for 24h. Three channels under different condition of inducer concentrations were shown, which also showed relative stable oscillations.

The authors should add an additional section describing and demonstrating a broadly important application. This will help demonstrate the broad utility of the newly designed circuit.

Reply: Thanks for the reviewer's comments and suggestions. We have added an additional section in discussion according to the Reviewer's comments.

“Of course, the modular repressilator will have many more specific applications such as the frequency analysis, frequency encoding, signal processing and so on, which was also mentioned above. Specially, one broadly important application should be mentioned, that is, pulsatile drug deliver system[23]. Because of the mechanical rhythms of our body, a continuous drug-release system are not ideal at some times, but a pulsatile drug deliver system, which can release the drug periodically. Although there were already some methods to realize the pulsatile drug deliver, they may all have some drawbacks. These kinds of methods were mainly physical methods, such as the coating layers or controlling plugs, need relatively complicated designs and the systems can only realize finite periods. In comparison, the modular repressilator can also be used to construct time controlled pulsatile drug system without the problems mentioned above. More importantly, the process of drug synthesis or stimuli generation may also be integrated into the engineering bacteria. So it may be an important future research direction to construct time controlled pulsatile drug deliver systems.”

Thank you again for your suggestions to improve the clarity of this paper.

Thank the reviewer again for all comments and suggestions.

Response to Reviewer #3

Reviewer Comments:

Reviewer #3:

General Comments: In this paper, Zhang et al. created a new oscillator circuit that can be orthogonally controlled by two inducers in Escherichia coli. By systematically tuning the concentration of the inducers, the amplitude and period can be modulated independently. They also constructed a quantitative model to forecast the regulation results. Under the guidance of the model, the expected oscillation can be regulated by choosing the proper concentration combinations of inducers. This novel design allows

the oscillator to be modularized and used in more complex circuit designs, which would be a valuable contribution to the Synthetic Biology field.

Reply: We thank the reviewer for the thorough and positive assessment of our work on the “This novel design allows the oscillator to be modularized and used in more complex circuit designs, which would be a valuable contribution to the Synthetic Biology field”. We have revised our manuscript following the reviewer’s suggestions to make the manuscript clearer and more rigorous.

The main critique I have is that I think the impact of the project needs to be explained more in detail. In line 52-54, it says. “Because this is a complete synthetic circuit unlike those in the natural world, it is difficult to control the oscillation in the absence of a reference.” This statement seems to be the one that justifies the circuit design but needs to be clarified, what is different in this synthetic circuit from natural systems? Why is this needed and important?

Reply: We gratefully appreciate the reviewer’s valuable comments and suggestions. We have re-written the paragraph according to the reviewer’s suggestion.

“Many important physiological processes are controlled by oscillatory signals, such as the P53 protein, cell cycles [1][2] and biological circadian rhythms [3][4]. Via complex control circuits [5], organisms can achieve stable oscillations and adjust the correlation parameters—period, amplitude and phase—accurately. Compared to these natural cyclic phenomena, synthetic oscillations are difficult to control in such complex networks for now; thus, stable and accurate control of oscillation is a problem. It is needed to be emphasized synthetic circuit with the independent control of amplitude and period is important because of its wide range of applications. The precise control can be quite useful for frequency analysis of downstream networks, frequency encoding and pulse-based signal processing. A modularized regulatable synthetic oscillation circuit can also be used in the design of more complex gene circuits, such as two-dimensional biosensors who need distinct responses from amplitude and period, periodic administration of therapeutic molecules[6].”

Recently, synthetic biology has sought to make oscillations robust and tunable in two

kinds of synthetic oscillators: the dual-feedback oscillator (DFO) [7] and repressilator (RLT) [8]. For the DFO, some amount of control has been achieved using two changed inducer concentrations [9] or extra control components [10]. However, the independent control of the RLT, which was developed earlier and is more widely researched than the DFO, has still not been realized. Based on its general principle, that is, a negative feedback loop with a time-delay [11], the repressilator(RLT) is constructed of three transcriptional repressors acting in sequence [9]. The three transcriptional repressors contain TetR from the Tn10 transposon, cI from bacteriophage λ and LacI from the lactose operon. In a period of oscillation, each repressor inhibits the transcription of the next one (a-b-c-a), which gives a time delay and leads to the oscillation. However, for now, synthetic oscillation circuits, such as the synthetic repressilator, unlike which the control of period and amplitude had been realized in the natural world, is difficult to control the oscillation in the absence. Two recent studies have reported significant progress: improvement of stability [12] and the theoretical direction of orthogonal control [6]. These works set the stage for independent controlling repressilators experimentally.”

Please check for typos:

Fig 4A. It should be “Amplitude”.

Reply:

Thanks for the reviewer’s careful check. We are very sorry for our negligence of some spelling mistakes. We have made corrections according to the Reviewer’ s comments.

Fig 1A. If Input I1 regulates the period and Input I2 regulates the amplitude why in the figure the schematic at the top left corner of each box shows the opposite?

Reply: Thanks for the reviewer’s careful check. We are very sorry for our negligence of the rationality of picture arrangement. We have redrawn the Fig.1A according to the Reviewer’ s comments.

Thank you again for your suggestions to improve the clarity of this paper.

Line 158, “and the independent regulation respectively”. Please complete the sentence, there’s missing information there.

Reply: Thanks for the reviewer’s careful check. We are very sorry for the mistakes in our language. We have added the missing information “Fig. 4b represented the independent regulation of period” according to the reviewer’s suggestion. Thank you again for your suggestions to improve the clarity of this paper.

Other comments/questions:

Lines 179-184. Upper limit is mentioned and explained in a nice way there, it may be worth informing the lower limit to complete the range in which the system is functional.

Reply: Thanks for the reviewer’s suggestion. We considered about the informing of the lower limit and did some experiments for supplement. To inform the lower limit of

amplitude, we thought the existing results were not enough because the gap of amplitude between the 0 μ M salicylate group and 50 μ M salicylate group (about 550a.u.) was too big compare with the noise level (about 10a.u.). So we added four groups of salicylate concentrations (0 μ M, 6.25 μ M, 12.5 μ M, 25 μ M) and the results were as following. The average amplitude of the 0 μ M salicylate group was 16.7a.u. and the average amplitude of the 6.25 μ M salicylate group was 48.0 a.u.. The 0 μ M salicylate group was just considered as the result of promoter leak and it's quite small comparing to the noise (about 10a.u.) so we couldn't inform that the amplitude was regulated. Thus we can inform that the amplitude range can be controlled from 48.0 ± 9.4 a.u. to 1948.8 ± 16.6 a.u.. As for the period, the gap of amplitude between the 0 μ M IPTG group and 2.5 μ M IPTG group (about 20min) was small enough comparing to the base noise value of the period (about 4min), so we thought there is no need to add extra experiments. The average period of the 0 μ M IPTG group was 323.9min. This group was significative because it represent the base period of the repressilator and it can't be regulated down in our design. So we think it's appropriate to set 323.9min as the lower limit of the period, and it can be informed that the period range can be controlled from 323.9 ± 3.3 min to 383.1 ± 1.9 min. These contents have also been added into the main text and the supplementary. Thank you again for your suggestions to improve the clarity of this paper.

Line 189. Please inform the modifications introduced here to Tomazou's work.

Reply: Thanks for the reviewer's suggestion. We are very sorry for our negligence to inform the detail of the modifications in the model section. We have rearrange the sentences and added the details of the modifications into the paragraph.

“To systematically describe independent regulation in development, we constructed a quantitative model. The model was mainly based on Tomazou's work [11] with some modifications, which contain changing a regulation of an additional node into an inducer and removing the enzymatic degradation.”

Thank you again for your suggestions to improve the clarity of this paper.

Lines 212-217. Why choosing the approach of removing the enzymatic degradation altogether when in Tomazou's paper the method is mentioned but not used?

Reply: Thanks for the reviewer's insightful comments. Here we can give an explanation that Tomazou's paper was an absolute theoretical article. On the theoretical level, although the method of introducing an extra enzymatic degradation system and removing the enzymatic degradation altogether can both achieve the decoupling of amplitude and period, the former have more advantages. By setting the parameters of the extra enzymatic degradation system, the circuit can be regulated more flexibly. Besides, removing the enzymatic degradation altogether means that relevant proteins can only dilute with the cells dividing. That may lead to an overall slowing down of the oscillation. So there was no need to use a relatively poor method in an absolute theoretical article. However, the condition was completely opposite on the experimental level. Introducing an extra enzymatic degradation system is quite difficult and may have some unpredictable effects on the experimental level, but removing the enzymatic degradation altogether is much easier and reliable. So we chose the latter method in our work.

For clarification, please elaborate what is different from the other papers mentioned, so that the reader doesn't need to read the other two papers to understand the point made here:

Lines 218-219, "simpler than Tomazou's method", in what way?

Line 276, "Our work profited much from Potvin's work", how?

Lines 279-281, "The method we used to regulate the period is a simplification compared to the design in Tomazou's work.", in what way?

Reply: Thanks for the reviewer's comments and suggestions. We are very sorry for neglecting to elaborate the differences between our work and the mentioned papers in detail. We have added the related instructions according to the Reviewer's comments. "For period regulation, we used IPTG to weaken the repression of LacI. This is simpler than the method to add an extra activator to regulate the expression of TetR, which was

used in Tomazou's method."

"Our work profited much from Potvin's work [10], which created highly regular and robust oscillations and presented the idea of simply removing existing features. Stability is the basis of regulation. Another important point is the choice of components. Quantifiable gene components are conducive to realizing regulation. Last but not least, it is important to appropriately simplify the gene circuit without impacting its function. The method we used to regulate the period is to weaken the repression of one node by inducer, which is a simplification compared to the design of an extra activator node used in Tomazou's work. Removing the enzymatic degradation altogether can be seen as a simplification."

Thank you again for your suggestions to improve the clarity of this paper.

In figure 4C. Why is it that the model predicts better the MI (I1, Amp) rather than MI (I2, Amp)?

Reply: Thanks for the reviewer's insightful comments. We analyzed the difference between MI (I1, Amp) and MI (I2, Amp) in experiment group and model group and found out that the main difference is the MI (I2, Amp) in model group is lower than MI (I2, Amp) in experiment group, which means I2 had some effect to amplitude in experiments. For this problem, we can give an explanation that the amplitude difference in different I2 groups is small in model but the amplitude had relatively obvious fluctuation in experiments, which was seemed as same correlation between I2 and amplitude in experiments. But in MI (I1, Amp), the range of I1 and amplitude were relatively wider and the correlation between I1 and amplitude did have some decline even with amplitude error, the experiment results were closer to model.

Please clarify the following expressions:

Lines 256-258. "It can be seen that the experimental results are generally quantitative and conform to the model, which means the model has a degree of predictability." What does "generally" mean here? I'd expect a reproducible result. Can you estimate the degree of predictability?

Explanation and Modification:

We gratefully appreciate the reviewer's valuable comments and suggestions. We are very sorry for neglecting to estimate the degree of predictability in detail. We calculated the Root Mean Squared Error (RMSE) of our model with experimental results, which was defined as:

$$\text{RMSE} = \sqrt{\frac{1}{m} \sum_{i=1}^m (y_i - \hat{y}_i)^2}$$

The results showed that the RMSE of the amplitudes was 97.0a.u., and the RMSE of the periods was 4.3min. These errors are within our acceptable range, so we considered that the model was predictive. These contents were also added into the main text, thank you again for your suggestions to improve the clarity of this paper.

Line 130. "The regulation also worked." What kind of regulation and in which sense it did work?

Line 131. "The fluorescence intensity amplitude, defined as the difference between a peak and the nearest previous trough, increased by approximately 100 times, generally after the addition of salicylate". What does "generally" mean here? I'd expect a reproducible result.

Reply: Thanks for the reviewer's comments and suggestions. We are very sorry for not giving a clear description of the regulation. We have rearrange the sentences according to the reviewer's suggestion.

"The average fluorescence intensity amplitude, defined as the difference between a peak and the nearest previous trough, increased from 17.9 a.u. to 1963.0 a.u. after the addition of salicylate, which means the regulation of amplitude worked".

Thank the reviewer again for all comments and suggestions.

References

- [1] Xue H, Shi H, Yu Z, He S, Liu S, Hou Y, Pan X, Wang H, Zheng P, Cui C, Viets H, Liang J, Zhang Y, Chen S, Zhang HM, Ouyang Q.(2014). Design, construction, and characterization of
- 20

a set of biosensors for aromatic compounds. *ACS Synth Biol.*3(12):1011-4.

[2] Zhang HM, Chen S, Shi H, Ji W, Zong Y, Ouyang Q, Lou C.(2016). Measurements of Gene Expression at Steady State Improve the Predictability of Part Assembly. *ACS Synth Biol.* 5(3):269-73.

[3] Zong, Y., Zhang, H. M., Lyu, C., Ji, X., Hou, J., Guo, X., ... & Lou, C. (2017). Insulated transcriptional elements enable precise design of genetic circuits. *Nature communications*, 8(1), 1-13.

[4] Rhodius VA, Segall-Shapiro TH, Sharon BD, Ghodasara A, Orlova E, Tabakh H, Burkhardt DH, Clancy K, Peterson TC, Gross CA, Voigt CA.(2013). Design of orthogonal genetic switches based on a crosstalk map of σ_s , anti- σ_s , and promoters. *Mol Syst Biol.*9:702.

[5] Wang, P., Robert, L., Pelletier, J., Dang, W. L., Taddei, F., Wright, A., & Jun, S. (2010). Robust growth of *Escherichia coli*. *Current biology*, 20(12), 1099-1103.

REVIEWERS' COMMENTS:

Reviewer #1 (Remarks to the Author):

I am satisfied with the revised ms. and the rebuttal to my earlier remarks. As far as I am concerned, the paper can fly now as it is.

Reviewer #2 (Remarks to the Author):

The revised manuscript is much improved and has addressed all my concerns. In particular, improving the motivation for the work and including statistical analysis. Nice work!

Reviewer #3 (Remarks to the Author):

Thanks for the thorough explanation of the impact of the whole project. I understand much better the goal of the current work now.

After going through all the responses I think that while the results presented here are impressive, the focus is very technical, as Reviewer 2 also commented. I agree that it would be more appropriate for a specialized journal, such as ACS

Synthetic Biology, in which the readers may want to use this system or the logic behind to build their own circuits.

I don't see how the results shown in this paper helps to advance an area of inquiry within the biological sciences, as the audience of Comm Biology would be interested in reading. I definitely believe this technology will be highly appreciated by the SynBio community.

Response to Reviewer #1

Reviewer Comments:

Reviewer #1:

General Comments:

I am satisfied with the revised ms. and the rebuttal to my earlier remarks. As far as I am concerned, the paper can fly now as it is.

Reply: We thank the reviewer for the thorough and positive assessment of our work. Thanks again for the suggestions to improve the article in the previous peer review.

Reviewer #2:

General Comments:

The revised manuscript is much improved and has addressed all my concerns. In particular, improving the motivation for the work and including statistical analysis. Nice work!

Reply: We thank the reviewer for the thorough and positive assessment of our work. Thanks again for the suggestions to improve the article in the previous peer review.

Reviewer #2:

General Comments:

Thanks for the thorough explanation of the impact of the whole project. I understand much better the goal of the current work now.

After going through all the responses I think that while the results presented here are impressive, the focus is very technical, as Reviewer 2 also commented. I agree that it would be more appropriate for a specialized journal, such as ACS

Synthetic Biology, in which the readers may want to use this system or the logic behind to build their own circuits.

I don't see how the results shown in this paper helps to advance an area of inquiry within the biological sciences, as the audience of Comm Biology would be interested in reading. I definitely believe this technology will be highly appreciated by the SynBio community.

Reply: We thank the reviewer for the thorough and positive assessment of our work. Here we want to explain the reason to choose *Communications Biology*. *Communications Biology* is an open access journal from Nature Portfolio publishing high-quality research, reviews and commentary in all areas of the biological sciences, of course including the synthetic biology. And it can be seen *Communications Biology* is great-hearted with the increasing articles and impact factor, so we chose to send our article to *Communications Biology*. Thanks again for the suggestions to improve the article in the previous peer review.